# OSNeRF: On-demand Semantic Neural Radiance Fields for Fast and Robust 3D Object Reconstruction

## ABSTRACT

By leveraging multi-view inputs to synthesize novel-view images, Neural Radiance Fields (NeRF) have emerged as a prominent technique in the realm of 3D object reconstruction. However, existing methods primarily focus on global scene reconstruction using large datasets, which necessitate substantial computational resources and impose high-quality requirements on input images. Nevertheless, in practical applications, users prioritize the 3D reconstruction results of on-demand specific object (OSO) based on their individual demands . Furthermore, the collected images transmitted through high-interference wireless environment (HIWE) leads to negatively impact the accuracy of NeRF reconstruction, thereby limiting its scalability. In this paper, we propose a novel on-demand Semantic Neural Radiance Fields (OSNeRF) scheme, which offers fast and robust 3D object reconstruction for diverse tasks. Within OSNeRF, semantic encoder is employed to extract core semantic features of OSOs from the collected scene images, semantic decoder is utilized to facilitate robust image recovery under HIWE conditions, lightweight renderer is employed for fast and efficient object reconstruction. Moreover, a semantic control unit (SCU) is introduced to guide above components, thereby enhancing the efficiency of reconstruction. Demonstrative experiments demonstrate that the proposed OSNeRF enables fast and robust object reconstruction in HIWE, surpassing the performance of state-of-the-art (SOTA) methods in terms of reconstruction quality.

## CCS CONCEPTS

• **Computing methodologies** → Computer graphics; Computer graphics..

## KEYWORDS

3D reconstruction, neural radiance field, semantic encoder and decoder, on-demand object, lightweight renderer

## 1 INTRODUCTION

Three-dimensional (3D) object reconstruction [1–3] stands as a pivotal challenge within the realm of computer vision [4–6]. The Neural Radiance Fields (NeRF) [7–9] has recently risen as an exciting technique, providing a novel way to tackle the task of 3D object reconstruction. NeRF is able to compress a scene into a learnable model given multiple images and corresponding camera poses of

*ACM MM, 2024, Melbourne, Australia*

© 2024 Copyright held by the owner/author(s). Publication rights licensed to ACM.

ACM ISBN 978-x-xxxx-xxxx-x/YY/MM

https://doi.org/10.1145/nnnnnnn.nnnnnnn

the scene [10]. By incorporating a volumetric rendering skill [11], images of unseen camera views can be generated with convincing quality. Existing studies [12–14] mainly focus on global scene reconstruction. Nonetheless, in practical application, users tend to be more concerned with the reconstruction results of on-demand specific object (OSO) [15]. Consequently, NeRF schemes for global scene reconstruction that lack of on-demand often have significant inefficiencies [16]. Moreover, the inputs of the existing NeRF-based methods are high-quality images from the datasets. However, in real-world applications, collected images often become distorted during transmission through high interference wireless environments (HIWE) [17], which significantly compromising the quality of object reconstruction [18].

In this paper, we propose a novel on-demand Semantic Neural Radiance Fields (OSNeRF) scheme for fast and robust 3D object reconstruction. As depicted in Fig. 1, initially, the Semantic Control Unit (SCU) directs the cooperative robots to conduct data collection, guided by the user demand indicator, to obtain a multi-view representation of the 3D scene. Subsequently, the semantic encoder sequentially performs semantic segmentation, semantic feature extraction, and semantic feature compression on OSO images. Following that, the semantic decoder reconstructs the OSO images based on the received compressed semantic features. Finally, the restored images, containing only the core semantic features, are fed into a lightweight renderer. This scheme significantly reduces the computational complexity of NeRF while enhancing the efficiency and robustness of 3D object reconstruction in HIWE. In summary, our contributions are as follows:

- A novel on-demand semantic neural radiance fields (OSNeRF) scheme is proposed, which can provide fast and robust 3D object reconstruction in HIWE. With OSNeRF, on-demand objects in the scene are selectively reconstructed according to the user's indicators.
- Technically, we implement a prototype system of OSNeRF, which consists of on-demand data collector, semantic encoder, semantic decoder, lightweight renderer, and semantic control unit. By filtering redundant information and providing semantic-level reconstruction guidance, the high efficiency of 3D object reconstruction can be achieved.
- Comparison experiments are conducted, the reconstruction results clearly indicate that OSNeRF outperforms existing state-of-the-art (SOTA) methods in terms of both pixel-level and semantic-level metrics. Furthermore, OSNeRF exhibits a distinct advantage on speed and robustness in HIWE.

## 2 RELATED WORK

To bring out the motivation of OSNeRF and highlight its superiority against existing methods, we provide a comprehensive investigation about traditional 3D reconstruction method and NeRF-based reconstruction method in this section.

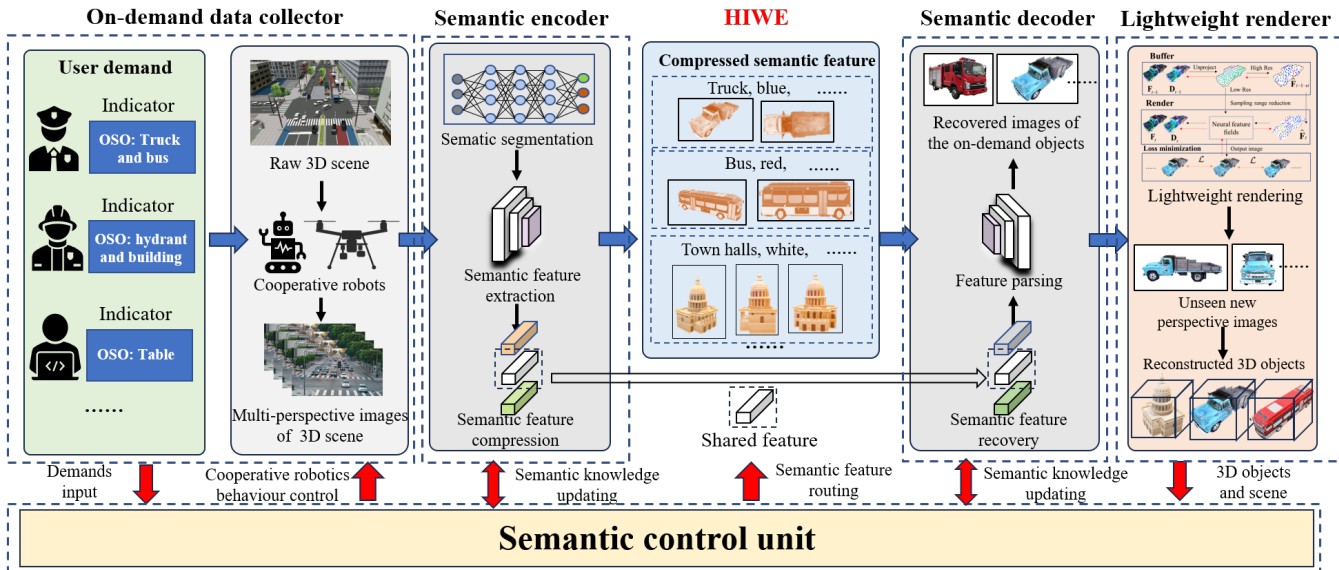

**Figure 1: The proposed OSNeRF pipeline. the collected images are initially processed by a semantic encoder, which compresses the semantic features for transmission in HIWE. Following this, the semantic decoder recover the images, which are then forwarded to a lightweight renderer designed to facilitate fast and robust 3D object reconstruction.**

## 2.1 Traditional 3D reconstruction

Traditional 3D reconstruction methods encompass both active and passive techniques, each offering distinct approaches to capturing spatial information [19]. In the active method, a structured light source is projected into the scene to determine target locations by extracting its projected information within the scene. the authors in [20] discuss the application of multi-sensor data fusion techniques with high accuracy indoor object modeling. To realize dynamic object reconstruction, [21] proposed an efficient direct tracking on the truncated signed distance function and leverage color information to estimate the pose of the sensor. Passive method utilizes ambient environmental cues, like natural light reflections, combined with images captured by cameras and analyzed through specific algorithms to generate 3D data, offering simplicity and high feasibility compared to active methods. Among these techniques, Photometric-Stereo facilitates the determination of normal vectors by utilizing stereo vision principles [22]. It achieves this by analyzing multiple images captured under varying lighting conditions but with consistent viewpoints. Similarly, the Shape From Shading (SFS) stands out in discerning surface orientations through meticulous examination of light and shadow variations [23]. Moreover, Multi-View Stereo (MVS) is a key method for recovering 3D structure by exploiting differences in the projected positions of the same 3D points observed by multiple cameras [24]. However, traditional 3D reconstruction methods are hampered by several limitations, including subpar performance in water environments [25] and with small objects [26], as well as substantial time that impede real-time reconstruction capabilities [27]. Traditional 3D reconstruction methods typically rely on image or point cloud data acquired from a limited number of viewpoints. This sampling restricts an accurate representation of the object, especially in occluded or detail-rich areas.

## 2.2 Neural radiance fields (NeRF)

Neural Radiance Fields (NeRF) represent a groundbreaking approach that learns the neural radiance field representation of a scene, enabling the synthesis of realistic novel views from limited 2D image observations. By modeling the geometry and appearance relationship of the scene, NeRF achieves high-quality reconstruction and rendering of complex 3D scenes. The remarkable performance of NeRF has inspired numerous extensions and explorations across diverse domains of 3D reconstruction. Notable extensions include human reconstruction [28–30], dynamic object reconstruction [31–35], and realization of reconstruction of large scenes [36–39], among others. These advancements showcase the versatility and potential of NeRF as a foundational framework. Nevertheless, as the complexity of the scene increases, the reconstruction process requires more hardware resources and time. To address this issue, some research efforts have focused on enhancing the rendering speed of NeRF. For instance, FastNeRF [40] introduces a novel light sampling strategy that dynamically adjusts the number of sampling points, leading to reduced repetitive calculations and improved model speed. Similarly, PlenOctrees [41] discretizes the continuous volume density and color function into a sparse octree structure, eliminating the need for redundant reasoning during real-time rendering. These methods primarily offer architectural improvements within the NeRF framework. It is important to note that the redundant information present in the input images can significantly impact reconstruction speed. Therefore, recent research has started exploring the utilization of semantic information in images to achieve more efficient reconstruction. A semantic-driven NeRF editing method is proposed in [42], which encodes texture editing in 3D space. Sem2NeRF [43] improves rendering accuracy

by encoding semantic masks into latent codes that control 3D object representation. Other methods, such as those presented in [13, 44, 45], integrate 3D space and semantic space modeling to enhance the model's ability in scene semantic editing and realistic rendering. However, it is worth noting that all of the aforementioned approaches do not consider the distortion of input images in high-interference wireless environments (HIWE), thereby lacking robustness in practical applications.

## 3 PROPOSED SCHEME

In this section, we will elaborate on the proposed OSNeRF as Figure 1, which comprises the following constituents: 1) On-demand data collector, 2) semantic encoder and decoder, 3) Lightweight renderer, and 4) semantic control unit.

### 3.1 On-demand data collector

The training of NeRF typically necessitates a substantial volume of input images and corresponding scene geometric information. However, for a particular demand, the user's attention may be solely directed towards reconstructing specific facets of the scene or particular vantage points. For example, firefighter's demand is to 3D reconstruct hydrants and buildings within risk scene, the on-demand data collector has the ability to selectively gather images pertaining to the hydrant and building objects present, and employ them as training data for the model. Consequently, the model's focus will be sharpened on acquiring a profound understanding of the objects' visual characteristics and structural attributes, thereby enhancing its performance on the reconstruction task. By employing a on-demand data collector, we can tailor the collection and utilization of data associated with a designated task, thereby augmenting the model's efficacy in relation to that specific demand. This methodology concurrently streamlines the scale and intricacy of the training data, whilst affording the model an opportunity to concentrate more intently on the pivotal aspects of the demand, thereby bolstering the efficiency of the training process.

### 3.2 Semantic encoder and decoder

The semantic encoder is deployed on the data transmitter to facilitate the transmission of compressed semantic features, thereby eliminating redundant information while preserving the quality of the 3D reconstruction. The encoding process involves semantic segmentation, semantic feature extraction, and semantic feature compression.

Initially, the collected raw images are normalized to enhance convergence speed and minimize computational loss. Subsequently, the normalized multi-perspective images $\mathbf{I_M}$ and on-demand indicators $\mathbf{O}$ are fed into fully convolution networks $F_\lambda$ for further processing. i.e.,

$$F_\lambda : (\mathbf{I_M}, \mathbf{O}) \rightarrow (\mathbf{M}, \alpha, \beta), \qquad (1)$$

where $\mathbf{M}$ symbolizes the generated mask with dimensions, each element indicates whether a pixel is included or excluded as part of the OSO (On-demand Semantic Object). $\alpha$ quantifies the degree of overlap between the mask and the actual annotation. $\beta$ represents the category label of the OSO. Additionally, we can calculate the predicted category label as $\hat{\beta} = \gamma(\hat{\mathbf{M}}, \hat{\alpha}, \mathbf{O})$, where $\gamma(\cdot)$ is a function that generates class label predictions based on the indicator $\mathbf{O}$, we

accomplish semantic segmentation and obtain the desired OSO through the training process.

Subsequently, the Swin Transformer [46] is utilized to extract semantic features from the input images, which are segmented into uniform patches via patch partitioning and linear embedding. These patches are sequentially processed through multiple layers of Transformer modules, each incorporating a window-based local attention mechanism. This mechanism selectively focuses on interactions among adjacent blocks, thereby decreasing both computational and memory complexities. The attention formula can be expressed as follows

$$\text{Attention}(\mathbf{B}, \mathbf{L}, \mathbf{Z}) = \text{Softmax}\left(\frac{\mathbf{B}\mathbf{L}^\top}{\sqrt{\rho}}\right)\mathbf{Z}, \qquad (2)$$

where Softmax($\cdot$) denotes an activation function. $\mathbf{B}, \mathbf{L}, \mathbf{Z}$ are the input embeddings obtained by linear transformation, $\rho$ is the adjustment factor. The semantic features extracted from the input images are represented as $(f_1, f_2, ..., f_l)$, where $l$ denotes the total number of semantic features.

To enhance the adaptability of semantic feature transmission across varying demands in high interference wireless environments (HIWE), we have developed a semantic-aware method that generates a feature weight vector based on both the user demand and the input data. This vector quantifies the importance of each semantic feature relative to the user demand. In our semantic-aware approach, we employ Grad-CAM [47] to produce a heat map. The classifier's output probability vector is represented as $\mathbf{g} = [g_1, \ldots, g_l, \ldots, g_c]$, the partial derivative feature vector (weight vector) $i_l$ is obtained by $i_l = \frac{\partial g_l}{\partial f_l}$, where the gradient information reveals the sensitivity of the feature vector $f_l$ to the user demand. A higher value of $i_l \in [0, 1]$ indicates greater importance of the feature for fulfilling user demand. The feature weight vector of all the $K$ user demands is obtained as follows

$$\mathbf{w} = \lambda_1 i_{l,1} + \ldots + \lambda_j i_{l,j} + \ldots + \lambda_K i_{l,k}, \qquad (3)$$

where $\lambda_j$ denotes the weight of the $j$-th demand, $i_{l,j}$ is the sensitivity vector corresponding to the $j$-th demand. Subsequently, the mask layer can abandon the redundancy semantic features while preserving the core semantic features (CSF) with higher $\mathbf{w}$, which are most relevant to task demand and can be repressed by $(r_1, r_2, \ldots r_m)$ and $m$ is the total number of the CSF, thus achieving intelligent semantic feature compression.

Specifically, the semantic compression is responsible compressing the redundant semantic features in the full semantic feature. The redundant semantic features refers to the semantic feature that are easily predicted based on image or are useless for driving the object reconstruction task. During semantic compression, we train a feature shared by the transmitter and the receiver to represent redundant information. Notably, the semantic features most relevant to the NeRF task and will not be compressed during semantic routing. Although the above method is a lossy feature compression. However, due to the introduction of basic knowledge [48], the performance of OSNeRF remains unaffected by this process since the core semantic features for the reconstruction can be near-perfectly recovered in HIWE.

For robust wireless transmission in HIWE, the power normalization layer is used to map the compressed semantic feature to

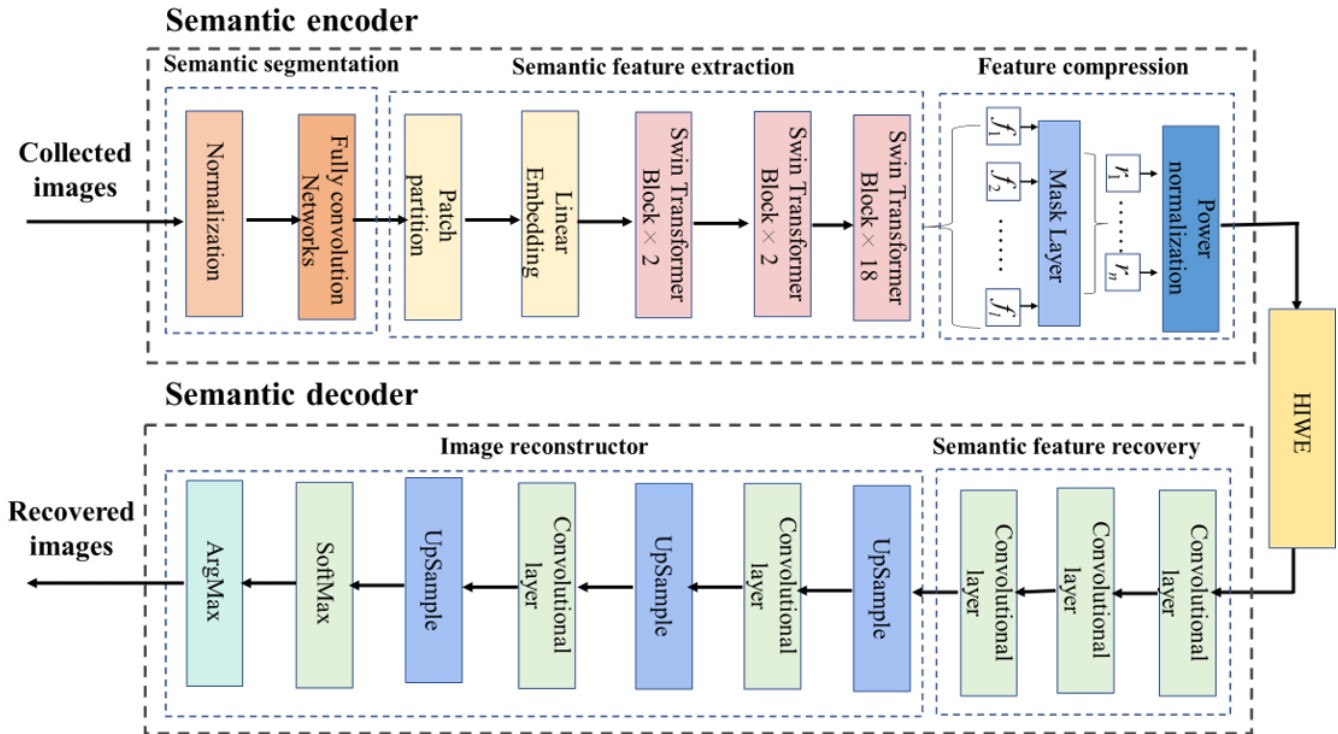

**Figure 2: The framework of the proposed semantic encoder and decoder of OSNeRF.**

the channel input sequences. During the training phase, a set of non-trainable layers are utilized to simulate widely-used wireless channel models, thereby enabling an end-to-end communication framework. Furthermore, a scalable semantic decoder is implemented in the physical layer, guided by the Semantic Control Unit (SCU) to facilitate adaptive decoding. The semantic recovery section comprises three convolutional layers designed to mitigate the impact of noise on the semantic features at the receiver. Each convolution layer incorporates several filters. The image reconstruction segment consists of three upsampling layers and two convolution layers, followed by a softmax activation function layer and an argmax layer. The upsampling layers are tasked with progressively restoring the original dimensions of the image, while the convolution layers sequentially extract semantic information. Ultimately, the argmax operation maps the recovered semantic features back to the original data space, thus preparing the reconstructed images to drive the subsequent rendering task.

### 3.3 Lightweight renderer

We obtain the reconstructed images from the semantic decoder, which are subsequently inputted into the lightweight renderer. Each pixel in the image is associated with a ray $\mathbf{r}(t)$ originating from the camera, determined by the camera parameters and defined as $\mathbf{r}(t) = \mathbf{o} + t\mathbf{d}$. Here, $\mathbf{o}$ represents the origin of the light source (i.e., the camera's position), $t$ is the parameter along the ray, expressed as a scalar, and $\mathbf{d}$ is the direction of the ray corresponding to the pixel. We sample multiple points along the ray and provide them, along

with their respective directions, as inputs to the neural network $F_\theta$. This allows for the prediction of both the color $\mathbf{c}_r$ and the depth $d_r$, which can be obtained as follows

$$\mathbf{c}_r = \sum_{i=1}^{N} T_r^i \alpha_r^i \mathbf{c}_r^i \quad d_r = \sum_{i=1}^{N} T_r^i \alpha_r^i t_r^i, \tag{4}$$

where $N$ denotes the number of samples sampled uniformly between the near and far planes, $\alpha_r^i = 1 - \exp\left(-\sigma_r^i \delta_r^i\right)$ and $T_r^i = \prod_{j=1}^{i-1} \left(1 - \alpha_j\right)$ denote the transmittance and alpha value of each sampled point, respectively. Subsequently, we employ volume rendering [49] to generate a feature map, which serves as a neural network approximation of the radiance field. This representation captures the color and volume densities at each point and in each viewing direction within the scene. Consequently, the static object is effectively modeled as a continuous vector function, denoted as

$$F_\theta : \left(\mathbf{x} \in \mathbb{R}^3, \mathbf{d} \in \mathbb{S}^2\right) \mapsto \left(\sigma \in \mathbb{R}, \mathbf{f} \in \mathbb{R}^K\right), \tag{5}$$

where $\mathbf{x} = (x, y, z)$ denotes the the spatial coordinates of a point within a three-dimensional space. $\mathbf{d}$ denotes the observation direction. $K$ represents the number of channels within our feature vector. Through the process of volume rendering, we calculate the feature vector for each ray $r$ using the equation $\mathbf{f}_r = \sum_{i=1}^{N} T_r^i \alpha_r^i \mathbf{f}_r^i$. By selectively rendering the feature vectors for a subset of rays, we generate a feature map denoted as $\mathbf{F}$. Additionally, we render a low-resolution depth value, denoted as $\mathbf{D}$. Both $\mathbf{F}$ and $\mathbf{D}$ are stored in buffer for subsequent optimization.

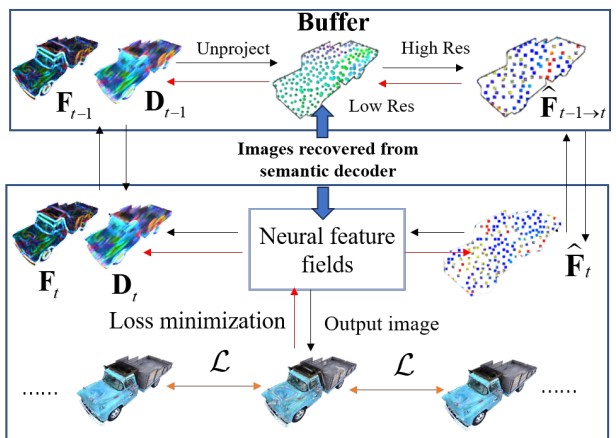

**Figure 3: The principles of lightweight renderer, the inputs are the images recovered from the semantic decoder. The buffer saves previous feature map and depth map, which can be used to accelerate rendering at the current viewpoint.**

By utilizing the buffer, which contains the previous $L$ feature maps $\{\mathbf{F}_{t-L}, \ldots, \mathbf{F}_{t-1}\}$ and the depth map $\{\mathbf{D}_{t-L}, \ldots, \mathbf{D}_{t-1}\}$, we leverage the stored semantic information to guide the selection process for determining the current sampling position. This approach significantly enhances the rendering speed. Specifically, by combining the current viewpoint's feature map with the maps in the buffer, we can generate low-resolution feature maps. Additionally, we employ a strategy similar to [9] to further reduce the number of sampling points, resulting in faster rendering of the generated low-resolution feature maps.

We subsequently project the 3D point cloud directly onto a high-resolution to generate feature maps that boast enhanced resolution and precision. i.e.,

$$\hat{\mathbf{F}}_{t'\to t}\left(\lfloor \hat{u}_{t'\to t}\rfloor, \lfloor \hat{u}_{t'\to t}\rfloor\right) = \mathbf{F}_{t'}(u, v), \tag{6}$$

Subsequently, the reprojected high-resolution feature maps $\{\hat{\mathbf{F}}_{t'\to t}\}$ are connected to the up-sampled feature maps $\hat{\mathbf{F}}_t$ and mapped onto the output multi-view images:

$$g_\theta : \left(\{\hat{\mathbf{F}}_{t'\to t}\}, \hat{\mathbf{F}}_t\right) \mapsto \mathbf{I}_t, \tag{7}$$

As shown in figure 3, the core concept behind the proposed lightweight renderer is to optimize image rendering from the current viewpoint by utilizing previously restored low-resolution features and depth information stored in a buffer. The efficient renderer utilizes the reprojected feature maps at a higher resolution, along with the upsampled feature map, to generate the final image. After the low-resolution feature rendering, sample range optimization, and reprojection of previous frames, we proceed to minimize the loss function by contrasting the color of output image with the input image, which can be expressed as

$$\mathcal{L}(\theta) = \sum_{m_1=1}^{M_1} \sum_{m_2=1}^{M_2} \left\| C_{m_1}\left(\mathbf{r}_{m_2}\right) - \hat{C}_{m_1}\left(\mathbf{r}_{m_2}\right) \right\|^2, \tag{8}$$

where $M_1$ represents the number of images, while $M_2$ corresponds to the quantity of pixels contained within each image, $C_{m_1}\left(\mathbf{r}_{m_2}\right)$ and $\hat{C}_{m_1}\left(\mathbf{r}_{m_2}\right)$ denote the true color and predicted color of the $m_2$-th pixel on the $m_1$-th image, respectively. By implementing optimization methods such as the Stochastic Gradient Descent (SGD) algorithm [50], we can progressively update the parameters $\sigma$ pertaining to the neural network $F_\theta$. In practice, we have improved the U-net [51] neural renderer by increasing the number of low-resolution feature convolution layers while decreasing the number of high-resolution feature convolution layers, thus substantially reducing the rendering time while ensuring the visual quality. These iterative updates are intended to minimize the OSNeRF loss function and ultimately achieve accurate object reconstruction.

### 3.4 Semantic control unit

The semantic control unit (SCU) is the semantic information interaction center of our proposed OSNeRF. Its main functions can be summarised as follows.

**Cooperative robots behaviour control.** The SCU has the ability to convert the user's demand specifications into associated semantic data, thus directing the behavior of the cooperative robots (drones, intelligent vehicle, robot dogs, etc.) towards capturing the raw images most relevant to fulfilling the task requirements. This enhances the dependability of the operational cooperative robots and heightens the efficiency of the data collection endeavor.

**Semantic knowledge updating.** The update of semantic knowledge guarantees the preservation of consistent semantic comprehension between the encoder and decoder. Updating and promptly disseminating the most recent semantic knowledge and concepts help evade misunderstandings and disparities in the coding and decoding of semantic information. Furthermore, by updating the semantic knowledge base in situations where multiple OSNeRF systems share a common semantic repository, it ensures that these systems can comprehensively understand and interpret one another's information, thereby enriching collaborative endeavors and integration capabilities.

**Semantic feature routing.** Compressed semantic feature routing efficiently transmits compressed semantic information to the respective decoder and renderer, employing an effective routing mechanism. Furthermore, it employs compressed semantic features to steer the model's selection of scene regions of interest, thereby precisely allocating computational resources. By concentrating computational resources on regions of elevated semantic significance, such as objects or areas of interest, the reconstruction results are enhanced in terms of both quality and efficiency.

## 4 IMPLEMENTATION DETAILS

### 4.1 Dataset and baseline

We train our framework on the Tandt [52] dataset (composed of 251 images of 980 × 545 px). We partition the data to 200 training scenes and 51 testing scenes. We also test our model (merely trained on Tandt) on the ABO datasets [53], which diverse geometries with realistic materials. Moreover, in order to verify the generalizability of our proposed method, we collected a set of object images of real scenes by UAVs and robots and named them as Fyts dataset, which composed of 180 images of 1920 × 1080 px and diverse sharp and

intricate textures. The above datasets have different scene and view distributions from our training dataset. Models are trained on all images of the training set for 1M iterations.

For performance bench-marking, we compare the reconstruction results of the proposed OSNeRF with the SOTA schemes proposed in [13, 43, 54, 55], in which [54] and [55] are classical NeRF methods while [43] and [13] are semantic-based NeRF methods. To conduct a comprehensive analysis of multiple NeRF-based methods, the Nerfstudio framework is used as it incorporates multiple neural implicit surface reconstruction approaches into a single framework. The NeRF training was executed using a Nvidia 4090 GPU, while the geometric comparisons of the 3D results were performed on a standard PC.

## 4.2 Prototype system

OSNeRF focuses on implementing a robust and effective 3D object reconstruction in high-interference wireless environment. In particular, we will focus on the transmission of compressed semantic features. The SCU has the goal of enabling the decoder to perform inference on new data samples by exploiting a semantic paradigm. The hardware implementation will equip terminals with Jetson Nano processors for training and inference of large AI/ML models, softwaredefined radio (SDR) units provide a robust and effective implementation tested under various SNR wireless environment.

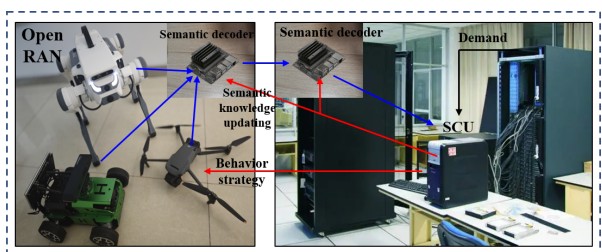

Figure 4: Experimental hardware setting. The on-demand data collector and semantic encoder are deployed at the transmitter side, and the remaining components are deployed at the receiver side. The blue arrows segments indicate the direction of data flow and the red arrows segments indicate the control commands of the SCU.

The prototype system and hardware setting are illustrated in Figure. 4. Multiple cooperative robots (drones, intelligent vehicle, robot dogs) are utilized for data collection. The connectivity layer for the robotics will be provided by an advanced 3GPPcompliant core network (Rel-17/18) and an advanced 5.5G Open RAN system [56] equipped with semantic awareness platform. The high interference wireless environment is modeled as a standard Rayleigh fading channel (Signal-to-noise ratio below 10dB). SCU establish a comprehensive semantic knowledge base for managing the semantic codec and robotics, the semantic model training in above devices will be integrated into the robots tested for large-scale deployment.

## 4.3 Evaluation metrics

**Pixel-level evaluation.** We utilize the evaluation methodology proposed in [57], where we initially capture multiple images of

the reconstructed object from identical viewpoints. Subsequently, we analyze the pixel-level differences between the images of the original 3D object and those reconstructed from corresponding perspectives. To quantify the evaluation, we employ a comprehensive set of metrics, including PSNR (Peak Signal-to-Noise Ratio), SSIM (Structural Similarity), LPIPS (Learned Perceptual Image Patch Similarity), and FID (Fréchet Inception Distance).

**Semantic-level evaluation.** Given the limitations of pixel-level metrics in assessing the semantic matching of reconstructed images, we propose a semantic-level evaluation to measure the performance of the generated images. To achieve this, we employ BLIP [58], a powerful visual language model that integrates visual language understanding and construction. This model enables us to convert the multi-perspective images into textual representations. Subsequently, we utilize large language models like BERT [59] to obtain embeddings of these generated texts. Finally, we compare the differences between the embeddings using cosine similarity (CS) and BLEU [60], the BLEU in this paper can be calculated as

$$\log \text{BLEU} = \min\left(1 - \frac{l_{\hat{s}}}{l_s}, 0\right) + \sum_{n=1}^{N} u_n \log \frac{\sum_k \min\left(C_k(\hat{s}), C_k(s)\right)}{\sum_k \min\left(C_k(\hat{s})\right)}, \quad (9)$$

where $n$-grams means that the size of a word group. $s$ is the transmitted sentence with length $l_s$ and $\hat{s}$ is the decoded sentence with length $l_s$, $C_k(\cdot)$ is the frequency count function for the $k$-th elements in $n$-th grams.

## 5 EXPERIMENTS

The experiments follow the evaluation framework presented in [61], where we initially capture multiple images from identical viewpoints in both the processed 3D and recovered scenes. Subsequently, a meticulous analysis is conducted to assess pixel-level disparities between the images derived from the original 3D scene and those obtained from the reconstructed 3D scene, all captured from corresponding perspectives. For performance bench-marking, we compare the reconstruction results of the proposed OSNeRF with the SOTA schemes proposed in [13, 43, 54, 55]. To achieve fair and accurate comparisons, we run our method on the same experiment settings with other methods, and we try our best to directly use the reported official quantitative results in these papers or use the official code to run the experiments. The visual comparisons are shown in Figure 5, the quantitative results are expressed in Table 1 and Table 2.

**Qualitative comparison.** Figure 5 showcases our reconstruction results, which demonstrate exceptional visual quality across various datasets. Utilizing input images sampled from each test scene, we apply guidance-finetuning to derive the triplane scene code and assess reconstruction quality based on previously unseen images. Despite being exclusively trained on the Tandt dataset, our model exhibits remarkable generalization to the ABO and Fyts datasets, which feature diverse scene and view distributions. Notably, OSNeRF produces more regular geometries compared to the slightly skewed and distorted shapes generated by MvsNeRF and GPNeRF. Additionally, OSNeRF excels in capturing sharp details and reflective materials. In contrast, the application of DSNeRF

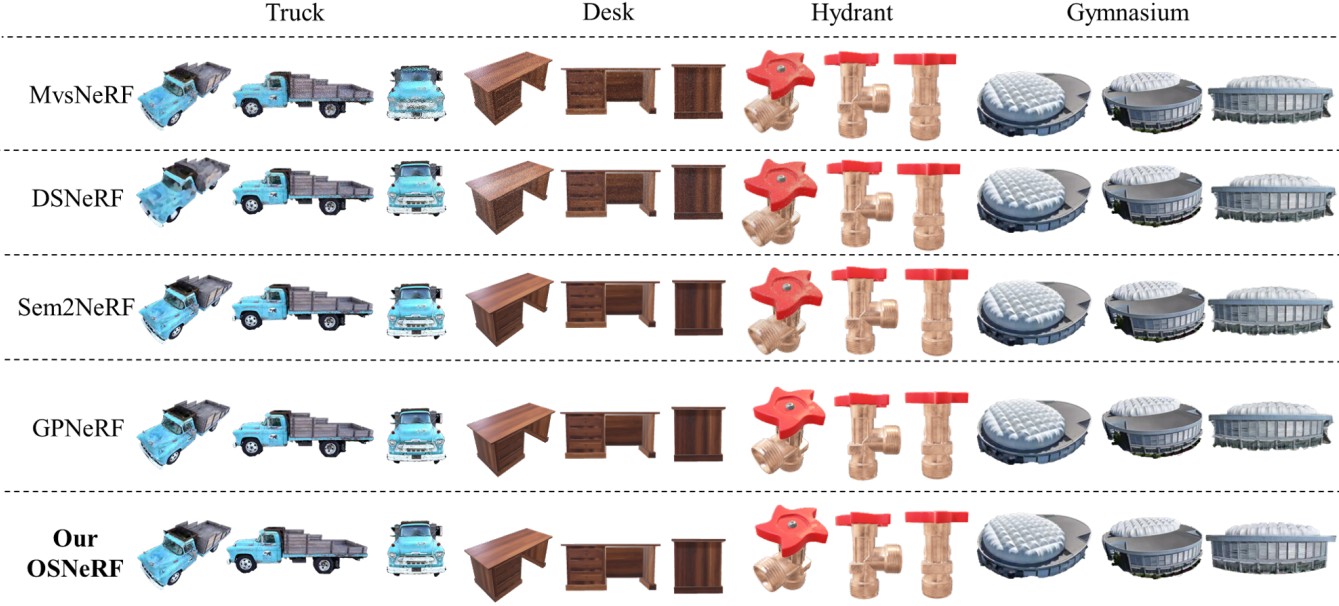

**Figure 5: Rendering quality comparison on the Tandt datasets (Truck)[52], ABO datasets (Desk) [53], and Fyts dataset (Hydrant and gymnasium) after 2h-processing. Note that the input images and semantic features of above NeRF methods are transmitted in HIWE (SNR=0dB)**

| Methods | Tandt [52] | | | | ABO [53] | | | | Fyts | | | |
|---|---|---|---|---|---|---|---|---|---|---|---|---|
| | PSNR ↑ | SSIM ↑ | LPIPS ↓ | FID ↓ | PSNR ↑ | SSIM ↑ | LPIPS ↓ | FID ↓ | PSNR ↑ | SSIM ↑ | LPIPS ↓ | FID ↓ |
| MvsNeRF | 24.07 | 0.825 | 0.092 | 29.95 | 23.44 | 0.770 | 0.189 | 33.95 | 21.18 | 0.658 | 0.252 | 35.95 |
| DSNeRF | 23.83 | 0.840 | 0.105 | 27.28 | 23.25 | 0.788 | 0.135 | 30.14 | 22.83 | 0.745 | 0.194 | 32.49 |
| Sem2NeRF | 25.86 | 0.892 | 0.087 | 23.65 | 24.52 | 0.837 | 0.108 | 27.91 | 23.36 | 0.791 | 0.155 | 29.08 |
| GPNeRF | 25.17 | 0.909 | 0.086 | 26.95 | 24.66 | 0.815 | 0.110 | 28.97 | 23.39 | 0.802 | 0.157 | 28.72 |
| **OSNeRF** | **26.61** | **0.926** | **0.078** | **21.39** | **25.97** | **0.919** | **0.084** | **23.58** | **25.12** | **0.903** | **0.096** | **24.27** |

**Table 1: The pixel-level quantitative comparisons of the various NeRF-based methods on the various datasets. Note that, our results achieve the best numbers in all four pixel-level metrics compared than other SOTA NeRF methods.**

| Methods | Tandt [52] | | ABO [53] | | Fyts | |
|---|---|---|---|---|---|---|
| | CS ↑ | BLEU↑ | CS ↑ | BLEU↑ | CS ↑ | BLEU↑ |
| MvsNeRF | 0.816 | 0.797 | 0.754 | 0.717 | 0.675 | 0.684 |
| DSNeRF | 0.889 | 0.804 | 0.877 | 0.822 | 0.716 | 0.715 |
| Sem2NeRF | 0.915 | 0.907 | 0.878 | 0.882 | 0.877 | 0.833 |
| GPNeRF | 0.904 | 0.885 | 0.892 | 0.895 | 0.908 | 0.845 |
| **OSNeRF** | **0.963** | **0.942** | **0.935** | **0.921** | **0.922** | **0.916** |

**Table 2: Quantitative comparisons based on the Semantic-level evaluation. Our OSNeRF similarly still remain higher CS and BLEU compared to other methods.**

to the ABO testing scenes results in noticeable blurring and tearing artifacts due to overfitting the training settings of the Tandit datasets. While the semantic-based methods (Sem2NeRF and GP-NeRF) outperform DSNeRF and MvsNeRF on the Fyts datasets in

actual scenes, both comparison methods exhibit flicker artifacts to varying degrees, more pronounced than those observed in our OSNeRF, as demonstrated in the appendix video. Consequently, we can conclude that OSNeRF achieves highly efficient and accurate 3D object reconstruction in HIWE environments.

**Quantitative comparison.** The pixel-level quantitative results are detailed in Table 1. While all methods achieve reasonable PSNRs, SSIMs, LPIPs, and FID on the Tandt testing set, our method consistently surpasses other methods across these four metrics when given the same input. More notably, our results on the additional two testing datasets significantly outperform the comparison methods, effectively demonstrating the robust generalizability of our OSNeRF. Typically, comparison methods aggregate 2D image features directly across view input at ray marching points for radiance field inference. In contrast, our method prioritizes semantic information and maintains the consistency of the reconstructed image

quality by leveraging the correlations among recovered semantic features. This leads to the best generalizablity and the highest rendering quality of OSNeRF across diverse testing scenes.

Table 2 showcases the semantic-level evaluation results on different datasets, further validating the performance of OSNeRF. The cosine similarity, which measures the semantic similarity between the original and reconstructed images, can reach up to 0.963. This demonstrates the high degree of semantic consistency between the two, indicating that OSNeRF successfully preserves the semantic information of the objects in the reconstruction. Furthermore, the BLEU score of OSNeRF reaches a maximum of 0.942, indicating a strong alignment between the original and reconstructed images at the semantic level, which suggests that OSNeRF effectively captures and preserves the semantic features of the original objects during the reconstruction process. Despite potential fluctuations in pixel values caused by variations in brightness, contrast, or color in the reconstructed images, the semantic consistency between the original and reconstructed images remains remarkably high. This indicates that OSNeRF effectively transmits the semantic features of the OSO while preserving semantic consistency, outperforming other methods.

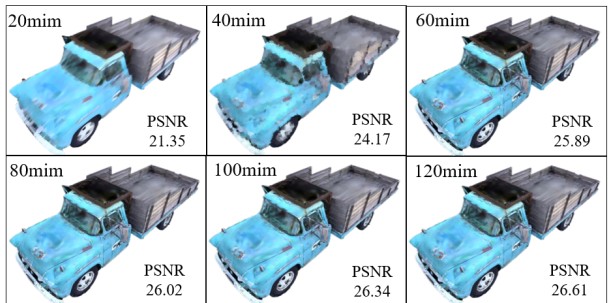

Figure 6: Optimization progress. We show results of our OS-NeRF construction result about truck with different time periods. The total processing time includes semantic coding and decoding time, data transmit time in HIWE, and image rendering time.

**Time comparison.** We present the 3D construction results of OSNeRF for a sample object (truck) with varying optimization durations in Figure 6. Notably, our reconstruction outcomes demonstrate substantial improvement within just 60 minutes of processing, compared to the state-of-the-art NeRF scheme depicted in Figure 5, which required 120 minutes for rendering. The visual quality of our reconstructed images is not only comparable but also superior. In addition, in the HIWE (SNR=0dB), the processing time required for our OSNeRF to generate an image with the same PSNR has been improved by more than 35% compared to the comparison schemes, the specific details are shown in the appendix video. This advantage arises because OSNeRF exclusively transmits the semantic characteristics of the object, rather than the entire original image. This approach significantly reduces both data transfer time and the processing load on the semantic decoder. Moreover, the semantic decoder exclusively restores the multi-view image encompassing the object, enabling the lightweight renderer to efficiently utilize

limited computational resources to focus on the object. Additionally, the SCU ensures efficient data transmission across various processing stages, leading to fast 3D object reconstruction.

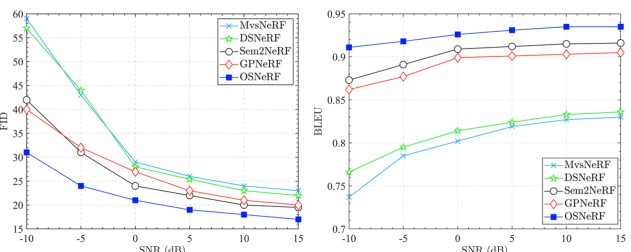

Figure 7: The quantitative comparisons (FID and BLEU) of various NeRF-based methods in different HIWE (-10dB to 15dB) after 2h-processing.

**Robustness comparison.** Figure. 7 illustrates the FID and BLEU of distinct NeRF techniques across HIWE with varying SNRs. It is evident that the reconstruction quality of OSNeRF surpasses that of the compared SOTA methods in high-interference wireless environments. This superiority arises from the distortion experienced by images transmitted directly through such environments, consequently affecting the object reconstruction outcomes. In contrast to the other SOTA methods, our OSNeRF incorporates a semantic encoder that accurately extracts the core semantic features of OSO based on user demand. Even in the presence of interference, the correlation between the key semantics through the SCU and the semantic decoder ensures consistency in both pixel-level and semantic-level details of the transmitted and received images. As the SNR improves, the performance of each scheme also improves. However, our OSNeRF consistently achieves the lowest FID and the highest BLEU across all SNRs, thereby highlighting the robustness of the proposed OSNeRF.

## 6 CONCLUSION

We present a novel On-demand Semantic Neural Radiance Fields (OSNeRF) scheme that can achieve fast and robust object construction in high interference wireless environment (HIWE). In contrast to traditional NeRF-based reconstruction, OSNeRF intensifies the focus on the semantic information of on-demand object (OSO). It incorporates an efficient on-demand data collector that procures multi-perspective images, employs a semantic encoder and decoder for precise feature extraction and robust image restoration in HIWE, and utilizes a lightweight renderer to expedite the reconstruction process. Additionally, we have developed a Semantic Control Unit (SCU) that orchestrates semantic-level services such as semantic routing for the above components. Experiments validates that the result of our OSNeRF performs favorably against state-of-the-art (SOTA) methods in terms of both both pixel-level and semantic-level, which enables fast and robust 3D object reconstruction in HIWE. For the future, we will further enhance our methodology to support real-time reconstruction of dynamic objects.

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
