# OpenReview forum: "OSNeRF: On-demand Semantic Neural Radiance Fields for Fast and Robust 3D Object Reconstruction"
_acmmm.org/ACMMM/2024/Conference — MM2024 Poster_

### Official Review · Reviewer_F7x5 · 2024-05-22

**Rating:** 2
**Confidence:** 4

**Summary:**

This paper proposes a novel on-demand Semantic Neural Radiance Fields (OSNeRF) scheme for fast and robust 3D object reconstruction. The key points are:

1. OSNeRF enables selective reconstruction of on-demand objects in a scene based on the user's requirements, rather than reconstructing the entire scene.

2. It employs a semantic encoder to extract core semantic features from the collected images and compress them for transmission in high-interference wireless environments

3. A semantic decoder is used to reconstruct the images containing just the core semantic features at the receiver end.

4. The reconstructed images are then fed into a lightweight renderer for fast and efficient 3D object reconstruction.

5. A semantic control unit guides the semantic encoder, decoder and lightweight renderer to enhance reconstruction efficiency.

6. Experiments show OSNeRF outperforms state-of-the-art methods in terms of reconstruction quality, speed and robustness, especially in high-interference wireless environments.

**Strengths:**

Proposes a novel on-demand semantic neural radiance fields (OSNeRF) scheme that focuses on reconstructing specific objects of interest rather than the entire scene. This on-demand capability is novel and practically useful.

Integrates semantic information to enable fast and robust 3D reconstruction of objects from images transmitted over interference-prone wireless channels. The use of semantics to improve reconstruction speed and robustness, especially in challenging wireless environments, is an innovative aspect.

Employs a well-designed pipeline consisting of semantic encoder, semantic decoder, lightweight renderer and semantic control unit. The semantic encoder extracts core semantic features and compresses them for robust wireless transmission. The decoder reconstructs clean images with core semantics for input to the lightweight renderer. This is a technically sound approach.

Leverages Swin Transformer for effective semantic feature extraction and Grad-CAM for identifying features most relevant to the reconstruction task. These are appropriate technical choices.

Conducts comparison experiments that demonstrate OSNeRF outperforms state-of-the-art methods on both pixel-level and semantic-level metrics. This provides good empirical evidence for OSNeRF's reconstruction quality.

Shows that OSNeRF has advantages in terms of speed and robustness, especially in high-interference wireless environments. Evaluating performance in the practically-relevant scenario of lossy wireless transmission is a strength.

Provides a clear description of the proposed OSNeRF scheme, elaborating on the key components like semantic encoder/decoder and lightweight renderer. The writing is generally clear and easy to follow.

**Limitations:**

The paper is well-structured and clearly written. I also enjoy reading the background and its applications.

However, this article has several serious problems.

1.The author claim that OSNeRF is compared with four SOTA schemes.
[1] GP-NeRF: Generalized perception NeRF for context-aware 3D scene understanding.
[2] Sem2nerf: Converting single-view semantic masks to neural radiance fields.
[3] Depth-supervised nerf: Fewer views and faster training for free.
[4] Mvsnerf: Fast generalizable radiance field reconstruction from multi-view stereo.

These papers are published at 2022. Considering how fast NeRF is growing in academia, they are definitely not the SOTA methods. Please compare with recent methods or methods published in at least 2023.

For instance:
[5] Simple-RF: Regularizing Sparse Input Radiance Fields with Simpler Solutions
[6] ColNeRF: Collaboration for Generalizable Sparse Input Neural Radiance Field
[7] Dngaussian: Optimizing sparse-view 3d gaussian radiance fields with global-local depth normalization
[8] Sparf: Neural radiance fields from sparse and noisy poses
[9] F2-nerf: Fast neural radiance field training with free camera trajectories

2.The proposed semantic encoder is a kind of neural feature fields, the proposed pipeline (e.g.  lightweight renderer in Fig.3) is not new in this area. The proposed advantages by the OSNeRF is a task of few-shot training, which is also not new in this area. Please give a literature review on these topics and highlight the differences between OSNeRF and existing methods.

**Suitability:**

2

---

### Official Review · Reviewer_B2ez · 2024-05-23

**Rating:** 4
**Confidence:** 2

**Summary:**

This paper proposed a novel on-demand Semantic Neural Radiance Fields (OSNeRF) scheme to resolve the transmission through the HIWE. This scheme structured with a SCU , a feature extraction encoder and lightweight render decoder. This scheme can improve the enhance the efficiency and robustness in reconstruction in HIWE.

**Strengths:**

1. Writing and paper organization is great
2. The proposed work represents a good idea for reconstruction and transmission between robots and servers. It is a novel idea to introduce this structured scheme to achieve the efficient transmission.

**Limitations:**

1. The field of “contribution to multimedia” looks like another paper, “TSNeRF,” instead of "OSNeRF."
2. The description of Figure 4 is not clear.
3. The dataset chooses to use why not use NeRF synthetic datasets
4. The training takes 1M iterations and the re-rendering cost 2h, is there any optimization method to improve these high system consumption cost?
5. The contrition claims this work can robust image recovery under HIWE conditions. Figure 6 and its explanation do not clearly explain the improvement under this condition.

**Suitability:**

3

---

### Official Review · Reviewer_zZeN · 2024-05-24

**Rating:** 2
**Confidence:** 4

**Summary:**

This paper proposed OSNeRF to enable 3D reconstruction of on-demand specific objects (OSOs), particularly in high-interference wireless environments. The authors introduced semantic encoding, semantic decoder, lightweight renderer, and semantic control unit for extracting semantic features, facilitating image recovery, reconstruction, and enhancing the efficiency, respectively. Experimental results demonstrate that the proposed OSNeRF outperforms some NeRF methods in terms of reconstruction quality.

**Strengths:**

The paper's organization is well-structured.

The authors have conducted a thorough evaluation using a diverse set of comparison datasets.

**Limitations:**

**Motivation**

The introduction lacks sufficient context and motivation for the proposed methodology. The paper transitions directly from the problem statement to the method description without adequately elaborating on the authors' rationale and the innovative aspects of the approach.
It would be helpful if the authors could provide a more detailed explanation for their choice of a semantic encoder-decoder architecture and how it compares to alternative techniques.


**Method**

1. While the authors present an architecture that combines a transformer encoder and a convolutional decoder for image reconstruction, the overall innovation of the method appears to be limited. The core process of encoding the image with a transformer, decoding it with a convolutional network, and then reconstructing the image using NeRF does not seem to significantly advance the state-of-the-art in this area.

2. The method description lacks the necessary clarity and specificity for a thorough understanding of the approach. For example, the distinction between the recovered RGB or semantic images in the semantic encoder-decoder module is unclear, which makes it difficult to assess the inputs and outputs of this critical component.

**Experiment**

1. While the authors have included several relevant baseline methods, the lack of details on how these approaches were adapted and configured for the current task limits the reader's ability to assess the validity of the comparative analysis. Specifically, the authors do not provide any information on how the generalizable GPNeRF method was modified to suit the OSO reconstruction task, which is necessary for a fair evaluation.

2. The choice of comparison methods appears to be somewhat outdated. It would be beneficial for the authors to consider incorporating more recent and state-of-the-art comparison methods, such as Instant-NGP [1] and Zip-NeRF [2].

[1] Instant neural graphics primitives with a multiresolution hash encoding. ACM transactions on graphics (TOG), 2022.

[2] Zip-nerf: Anti-aliased grid-based neural radiance fields. Proceedings of the IEEE/CVF International Conference on Computer Vision. 2023.

**Suitability:**

2

---

### Official Review · Reviewer_Pf3a · 2024-05-24

**Rating:** 4
**Confidence:** 2

**Summary:**

This paper introduces a novel approach to 3D object reconstruction.
This paper leverages semantic encoding and decoding to extract and transmit core semantic features of images, allowing for high-quality 3D reconstruction even in high-interference wireless environments. The approach incorporates a lightweight renderer and a semantic control unit (SCU) to significantly enhance reconstruction efficiency and quality, particularly in resource-constrained and high-interference scenarios.

**Strengths:**

1. The paper demonstrates that OSNeRF maintains high reconstruction quality even in high-interference wireless environments (HIWE).
2. OSNeRF introduces a Semantic Control Unit (SCU) to guide data collection and optimize the reconstruction process. This unit helps in enhancing the efficiency and accuracy of 3D object reconstruction by focusing computational resources on the most relevant parts of the scene as per user demand.
3. Notable performance improvement.

**Limitations:**

1. **Writing**: Citing each baseline method in results could enhance the writing.
2. **Potential Inefficiency in Feature Transmission**: A potential weakness of OSNeRF is that transmitting intermediate semantic features might not be as noise-resistant and data-efficient as directly transmitting compressed images to the local processing unit. Compressed images typically have better noise resilience and require less data to transmit. Therefore, the authors need to clarify the advantages of performing semantic segmentation directly on the robot side, such as any specific benefits in terms of robustness, efficiency, and overall system performance.

**Suitability:**

2

---

### Meta-Review · Area_Chair_TkYh · 2024-07-03

**Recommendation:** Accept (Poster)
**Confidence:** 5

**Metareview:**

This paper was reviewed by four experts in the field. The reviews are mixed, including Borderline Accept, Borderline Accept, Borderline Accept, Weak Reject. The authors have addressed majorities of the concerns from reviewers. The major concerns left by the reviewers are 1) lack of comparison on latest NeRF backbones, 2) the necessity of semantic information. While ACs agree with these two important concerns, we still think as an initial effort for an end-to-end controllable capturing and reconstruction system, this is still a valuable work. Moreover, additional experiments in rebuttal show the potential to generalize to other NeRF models, as well as the effectiveness of semantic information. Based on this, the decision is to recommend the paper for acceptance to ACM Multimedia 2024.

Still, as mentioned, we do think reviewers have raised valid concerns, and we recommend that the authors carefully read all reviewers' final feedback, and include and extend the experiments presented in rebuttal, to resolve those concerns. We congratulate the authors on the acceptance of their paper!